# Lung Transplant Rehabilitation—A Review

**DOI:** 10.3390/life13020506

**Published:** 2023-02-11

**Authors:** Yafet Abidi, Zsuzsanna Kovats, Aniko Bohacs, Monika Fekete, Saoussen Naas, Ildiko Madurka, Klara Torok, Levente Bogyo, Janos Tamas Varga

**Affiliations:** 1Department of Pulmonology, Semmelweis University, 1083 Budapest, Hungary; 2Department of Public Health, Faculty of Medicine, Semmelweis University, 1083 Budapest, Hungary; 3Department of Anesthesiology and Intensive Care, National Institute of Oncology, 1122 Budapest, Hungary; 4Department of Thoracic Surgery, National Institute of Oncology, 1122 Budapest, Hungary; 5Department of Pulmonary Rehabilitation, National Koranyi Institute of Pulmonology, 1122 Budapest, Hungary

**Keywords:** rehabilitation, lung transplant, exercise ability, survival, quality of life, exercise

## Abstract

Background: Both lung transplant recipients and candidates are characterised by reduced training capacity and low average quality of life (QoL). This review investigates the impact of training on exercise ability and QoL in patients before and after lung transplant. Methods: Searches were conducted from the beginning to 7 March 2022 using the terms “exercise,” “rehabilitation,” “lung transplant,” “exercise ability,” “survival,” “quality of life” and “telerehabilitation” in six databases, including Cochrane Central Register of Controlled Trials (CENTRAL), PubMed, CINAHL, Nursing and Allied Health, and Scopus. The inclusion criteria were studies evaluating the effects of an exercise training programme concurrent with lung transplantation as well as patients and candidates (>18 years old) through any lung diseases. The term “lung transplant rehabilitation” was used to refer to all carefully thought-out physical activities with the ultimate or intermediate objective of improving or maintaining physical health. Results: Out of 1422 articles, 10 clinical- and 3 telerehabilitation studies, candidates (n = 420) and recipients (n = 116) were related to the criteria and included in this review. The main outcome significantly improved in all studies. The 6-min walk distance, maximum exercise capacity, peak oxygen uptake, or endurance for constant load rate cycling improved measuring physical activity [aerobic exercises, breathing training, and aerobic and inspiratory muscle training sessions (IMT)]. Overall scores for dyspnoea improved after exercise training. Furthermore, health-related quality of life (HRQOL) also improved after aerobic exercise training, which was performed unsupervised or accompanied by breathing sessions. Aerobic training alone rather than combined with inspiratory muscle- (IMT) or breathing training enhanced exercise capacity. Conclusion: In conclusion, rehabilitation programmes seem to be beneficial to patients both preceding and following lung transplantation. More studies are required to determine the best training settings in terms of time scale, frequency, and work intensity in terms of improving exercise ability, dyspnoea, and HRQOL.

## 1. Introduction

Lung transplantation (LTx) has become a clinical reality for patients since the mid-1980s who suffer from end-stage lung disease. On the basis of statistics from the International Society for Heart and Lung Transplantation (ISHLT) database, the median patient survival among all indications was 6.7 years from 2010 to June 2017. The average survival for patients after the first year LTx is 8.9 years. Idiopathic interstitial pneumonia (IIP, 32.4%), chronic obstructive pulmonary disease (COPD, 26.1%), and cystic fibrosis (CF, 13.1%) are the predominant indications for LTx. Over the past decade, almost 4000 LTx have been reported worldwide each year [1]. To date, we are facing huge challenges, in particular, the imbalance of supply and demand among donors. This can result in an average waiting list of 326 days in the UK [2]. During this time, LTx candidates continue to suffer from medical chronicity, ventilatory restriction, training intolerance, and inactivity in everyday activities [2]. Quality of life (QoL) and peripheral and diaphragmatic muscle impairment are manifested [1,2,3]. Therefore, it is primordial to maintain a minimum daily functional capacity and prevent more complications [1,4,5]. Lung transplant rehabilitation is a highly recommended therapy for patients with long-term lung diseases who are simultaneously recovering after a transplant operation or candidates for LTx, [1,2]. There is a significant improvement in pulmonary functions after transplantation. However, the QoL is still limited by several factors, including early-onset metabolic acidosis, reduced daily exercise capacity (40–60%) of anticipated normal values, and skeletal muscle weakness [1,2,3]. In the early post-transplant period, prolonged hospital and intensive care unit (ICU) stays, which can last three to six weeks or longer if issues arise, result in prolonged inactivity and sedentary behaviour [1,2]. Furthermore, immunosuppressant medications continuously used after surgery have negative effects on the molecular structure of the skeletal muscle [1,2] and have a significant detrimental impact on HRQoL due to psychological stress [1,2,3]. When LTx patients and healthy people are compared one year after surgery, standing and walking time is dramatically reduced. However, daily sedentary time is still high [1,2,3], which might be partially explained by the malfunctioning peripheral muscles [3,4,5]. Quadriceps strength and muscle mass losses are also observed [1,2,3,4,5].

In these circumstances, exercise-based rehabilitation programmes should visibly improve the physical functioning, exercise tolerance, and levels of physical capacity of patients. In this review, the available data for rehabilitation interventions and the duration of hospitalisation will be investigated in the pre-transplant, early (≤12 months after surgery), and late (>1 year after emission regarding lung transplant procedure) post-transplant periods. Both evidence-based and empirical data will be proposed, and guidelines will be in focus for additional research.

## 2. Materials and Methods

From the beginning through 7 March 2022, searches were made in six databases, including the Cochrane Central Register of Controlled Trials (CENTRAL), PubMed, Scopus, Nursing and Allied Health, and CINAHL. These six databases were selected because they can be used in related systematic reviews and applied to clinical research. Medical topic headings (MeSH) relating to “rehabilitation,” “lung transplant,” “training exercise,” “exercise training capacity,” “quality of life,” “telerehabilitation.” and “survival” were the focus of the database-specific research methodologies.

As no translator was available, the search was limited to peer-reviewed articles in English. The authors separately evaluated the titles and abstracts to see whether the papers adhered to the predetermined PICOS (population, intervention, comparators, outcomes, and study design) standard. Lung transplant recipients and candidates (>18 years old) with any lung diseases met the inclusion criteria, and studies evaluating the effects of an exercise training regimen met the intervention criteria. Every planned and organised physical activity with the immediate or long-term goal of preserving or enhancing physical health was included in this definition.

A passive control group, an active training control group, or a specific dose, environment, or type of exercise were contrasted in randomised clinical trials (RCTs). They also looked into clinical outcomes (hospitalisations, length of stay in hospital or intensive care unit (ICU), and survival) as well as exercise training outcomes (6 min walk test (6MWT), endurance shuttle walk test (ESWT), incremental shuttle walk test (ISWT), or cardiopulmonary exercise test (CPET). General or respiratory questionnaires assessed the QoL, including HRQoL and psychological health.

## 3. Results

The six databases were searched, and 1,422 articles were found, mostly from PubMed/Medline (814), Cochrane Library (201), Nursing and Allied Health (172), Scopus (318), and CINAHL (89). After filtering full-text publications, 10 + 3 articles were found eligible and were included in the review. Figure 1 shows the PRISMA flow diagram of the article screening procedure.

Pretransplant rehabilitation studies

Pre-transplant candidates were included in eight studies out of ten (n = 420) with a median age of 51 years (ranging from 35 to 58 years) and an anticipated mean forced expiratory volume in 1 s (FEV_1_%pred) of 22 to 49%. Male applicants made up 43 to 95% of the study participants (median: 58%).

Pre-transplant studies included three randomized controlled trials (RCT) [2,3,4], two quasi-experimental studies [5,6], one cohort study, and two telerehabilitation research. Seven of eight trials [2,3,4,5,6,7] combined resistance and aerobic training, but only one [6] used Nordic Walking. Inpatient protocols were used in four research studies [2,4,6,8], whereas integrated inpatient and home-based exercise were used in two investigations [3,5]. The training sessions varied in frequency, with two to six sessions per week, and the duration of the programme was between four and twelve weeks (Table 1).

Post-transplant rehabilitation studies

Four out of the thirteen studies included post-transplant patients (n = 116), whose mean forced expiratory volume in 1 s (FEV_1_%pred) expected values varied from 73 to 81% and whose average age was 52 years (range from 47 to 58). Male participants made up 46 and 95% of the total population (mean: 61%). There were two RCTs [2,8], one quasi-experimental [11], one pilot research study [12], a and a telerehabilitation study (home-based pulmonary rehabilitation) [13] among the post-transplant investigations. One study looked at the impact of the HIIT programme [8], and four investigations were conducted as inpatient protocols [2,11,12]. Three to five training sessions were held each week, with training programmes lasting between three and twenty weeks (Table 2).

### 3.1. Results of Pre-Transplant Pulmonary Rehabilitation

The 6MWDT was used in all six personal pre-transplant research studies to assess exercise ability after the rehabilitation programme. The outcome significantly improved in five of the studies. However, Kerti et al. [2] found no appreciable change in handgrip strength (HGS) or 6MWD after the programme. Gloeckl R et al. [4] also looked into the distinction between interval and continuous training. They claimed that no differences existed between the two research groups, and both groups successfully increased the 6MWD by a homogenous substantial amount: 29–35 m for the interval training group and 36–43 m for the continuous training group. According to Ochman M et al. [6], Nordic walking increased the distance by 6MWT in a statistically significant way after a 12-week programme. The rehabilitation programme was combined with aerobic and resistance training in all six studies (Table 1).

QoL was evaluated in four pre-transplant studies, two of them [6,7] employed the SF-36 questionnaire, which produced two total scores and eight scales; the mental elements summary (MES) and the physical elements summary (PES), whereas one study [2] used the CAT questionnaire. Florian J et al. [5] examined the number of days spent on invasive mechanical ventilation (IMV) as well as the length of stay (LOS) in the hospital and the intensive care unit (ICU) after LTx in terms of survival following LTx. The SF-36 questionnaire was used in this study, but the findings ran counter to the LOS. After the PR programme, Fontoura F et al. [7] discovered that four key dimensions (vitality, emotional role, mental health, and physical functioning) of HRQL improved considerably. Ochman M et al. [6] found no statistically significant differences in QoL. According to Florian J et al. [5], the survival rate in the pulmonary rehabilitation programme (PRP) group (89.9%) was higher than in the control group (62.9%) during the follow-up period. The experimental group had a shorter hospital stay overall (20d PRP vs. 25d CG, *p* = 0.046) and a shorter ICU stay (five days vs. seven days, *p* = 0.004) than the control group. The PRP group had higher survival rates during the five-year follow-up period of lung transplantation (89.9% vs. 60.9%, with *p* < 0.001) and reported lower mortality rates during ICU stays (*p* = 0.006).

### 3.2. Results of Post-Transplant Pulmonary Rehabilitation

Five different tests were applied to evaluate the exercise capacity in the post-transplant group of patients: 6MWT [2,12], muscle strength hand and quadriceps power (1 RM) [8,11], peak oxygen uptake (VO_2_peak) [8], and endurance and incremental shuttle walk (ISWT, ESWT) tests [11]. Ulvestad M et al. [8] found that the mixed-mode HIIT, aerobic combined with resistance training 3 times/week for 20 weeks, improved muscular strength, although VO_2_peak did not improve rather to normal care after the operation. Candemir I et al. [11] reported significant improvement in endurance and incremental shuttle walk (ISWT, ESWT) tests, the distance of ISWT before PR (23%) of predicted to 36% following PR (*p* < 0.001), and the average score of quadriceps in one maximal repetition increased from 12 (6–17) kg to 13 (8–20) kg following PR with a significant difference (*p* < 0.001). Handgrip force in both hands (*p* < 0.05) improved after PR. For the 6MWD, both studies reported an increase in the distance of 6MWT. However, Andrianopoulos V et al. [12] reported a statistically significant difference (6-min walk test +86 m; *p* < 0.001), and Kerti M et al. [2] reported an improvement; however, it was not significant (Table 2).

QoL was evaluated in three out of four post-transplant studies. The following questionnaires were used: CAT [2], the SF-36 questionnaire [8], St George’s Respiratory function questionnaire, the Chronic Pulmonary Sickness questionnaire, Chronic Respiratory Sickness Questionnaire (CRQ), and the Hospital Anxiety Depression Scale (HADS) for psychological, mental health [11]. Neuropsychological testing for cognitive assessment was based on the Intelligence Quotient test (IQ) and the Stroop colour-word test [12]. Both Candemir I et al. [11] and Andrianopoulos V et al. [12] investigated the effect of the early post-transplant period two months after LTx operation and one month later. They reported statistically significant improvements in body composition, QoL, and psychological health status after the programme [11] and a statistically significant difference (ES latitude 0.23–1.00; *p* ≤ 0.34) in 50% of the cognitive exams measuring memory and learning capacity (ES scores 0.62 and 0.31, respectively), was detected [12]. Ulvestad M et al. [8] reported significant improvement in HRQoL in the late post-transplant from six to 60 months after LTx (Table 2).

## 4. Discussion

This study showed how training regimens affected physical function and improved QoL for patients with LTx. Although it has been hypothesised in the literature that a rehabilitation exercise training regimen can preserve or improve fitness capacity both before and after the transplant, the information currently available is restricted to non-randomised and observational research, and results have poor quality outcomes. The 6MWT is a crucial tool to assess the successful outcomes of the rehabilitation programme if the distance tends to exceed the minimal clinically relevant difference (MCID) set for chronic respiratory diseases [14,15]. The bulk of our studies shows that the training regimens have positive effects on physical function as well as enhancements in QoL for patients with LTx.

Kerti M et al. [2] reported a reduction in HGS following the transplantation and improvement after the rehabilitation training in FVC%pred, FEV_1_%pred, chest mobility, BHT, and QoL two months after the lung transplant. However, 6MWD remained at the same level in the post-transplant group. These results are similar to the results of Maury et al. [16]. In the study by Maury, the exercise training programme improved the functional components before the surgery, too. This study demonstrated significant improvement in the training capacity, chest wall expansion, and QoL. In a study by Kılıç L et al. [3], the average distance of 6MWT was 300 (70–524) meters (m) before the training and increased to an average of 360 m (60 m increase, *p* = 0.018) after the training protocol. The dyspnoea scale scores also improved significantly after the training. Dyspnoea is a major symptom of end-stage lung disease in people who are potential candidates for lung transplants [17]. PR has already been shown to reduce dyspnoea [18], and focusing on the diaphragm muscle during training improves the MRC dyspnoea scale significantly [19]. Thus, the average pre- and post-training MRC outcomes were statistically different (*p* = 0.008). The British Thoracic Society recommends a rehabilitation programme twice a week for six weeks under direct supervision [20]. However, some researchers found that a PR programme was ineffective twice a week for eight weeks [21]. In this programme, patients reported a reduction in muscle weakness and less pain at the end of the test alongside improvement in the distance in the 6-min walk test after rehabilitation. Current data suggested that pre-transplant rehabilitation could guarantee greater gains in the early stage [22]. Florian J et al. [5] are some of the few scientists who have looked into how a pre-transplant rehabilitation programme affects the survival and mortality of lung transplant patients. They found that 9.0% of PRP patients needed IMV longer than 24 h (compared to 41.6% in the control group), highlighting the crucial part that PRP plays in the recovery process after lung transplantation [23,24]. However, the prediction of mortality following LTx was not correlated with risk factors previously associated with early mortality, such as sex (male sex > 65), age (55), and one-lung transplant [25,26]. This conclusion can be explained by the fact that most of the individuals in this group were under the age of 65. Interval training versus continuous exercise for lung transplant patients was examined by Gloeckl R et al. [4]. Both interval and continuous training may result in equivalent increases in an exercise training capacity. These results are consistent with those of the earlier research findings by Puhan et al. [27]. In their study, there was no dissimilation within groups in terms of physical training or HRQoL, but both groups showed physiological benefits. The same research team showed in a second study that aerobic enzyme activity and the cross-sectional areas of type I and IIa fibers improved equally with IT and CT [28]. Leg fatigue and dyspnoea levels in the IT group were significantly lower than in the CT group. The CT group required significantly more breaks during the endurance workout. The number of breaks also rose over time, even though the IT group was able to keep the same frequency of pauses during training.

Following bilateral lung transplantation, Candemir I et al. [29] examined the effectiveness of outpatient pulmonary rehabilitation two months after the procedure. Body mass index (BMI) and fat-free mass index (FFMI) increased, according to his study. Baseline BMI was also connected with improvements in functional ability, and quadriceps muscular power was seen as a key indicator of an increase in muscle mass. Skeletal, muscular power, training potential, and QoL dramatically improved after rehabilitation. After PR, the ISWT distance improved from an expected 23% to a predicted 36%. Similar outcomes were shown in another trial when muscle strength and exercise duration increased in the early post-transplant period following a home-based training programme [30]. There is still some uncertainty over the best time for PR after lung transplantation. A randomised research study found that, compared to recipients who had no PR at all [31], lung transplant recipients who undertook an early PR programme (the first three months after surgery) had higher fitness, physical activity, and a decrease in 24 h blood pressure one year after LTx. In this study, the average postoperative time before PR was 75 +/− 15 days, which can be considered an early stage of rehabilitation.

Another review from 2010 reveals that skeletal muscle power, the functional and maximal ability for exercise, and QoL significantly improve in recipients following a training programme that began as early as the first month and lasted at least six months following LTx [29]. According to Andrianopoulos V et al. [12], lung transplant recipients with COPD as an underlying disease showed a number of benefits after participating in PR in functional as well as cognitive skills. Taking part in a three-week rehabilitation programme leads to multifactorial benefits in the early post-transplant period; it correlated with a faster and more convenient recovery one month after surgery. In this study, lung transplant recipients showed a significant amelioration of physical capacity evaluated by the 6MWT following a three-week rehabilitation programme. In terms of exercise tolerance, 6MWD (350 m) for COPD patients who underwent LTx due to a prior medical condition, poor health status, a lengthy hospital stay, as well as extended periods of inactivity additionally improved by 86 m following PR discharge. The findings of Munro et al. [32] indicated a gain of 95 m following a seven-week outpatient post-transplant PR, which is comparable to this large increase.

A study by Ulvestad M. et al. [8] examining the impact of peak oxygen uptake and muscle strength after intense exercise for six to sixty months following surgery discovered that HIIT significantly increased muscle power, which was accounted by an improvement of 11% in the 1RM leg press. The benefits of the HIIT programme seem to be the greatest for people who have had lung transplants, suggesting that HIIT helps hasten recovery. Langer et al. [31] evaluated the outcomes of a low- to moderate-intensity 12-week endurance and resistance training programme. No difference was observed in VO_2_peak in contrast to a control group receiving activity therapy in the intervention group. In earlier non-randomised controlled trials, exercise training significantly affected VO_2_peak [33,34]. These uncontrolled data, however, were unable to distinguish between the impacts of exercise training and the course of the normal recuperation of the body after LTx.

The SF-36 questionnaire, a universal tool for HRQoL evaluations, was the most frequently applied tool for QoL measurements [35]. It can be difficult to focus on changes in SF-36 scores when developing a basic diagnosis, although MCID for LTx candidates has not yet been established. A study shows a value of MCID of >2–4 points for PCS and MCS scores [35], and general recommendations call for an MCID of four points [34]. Lower training symptoms (dyspnoea and lower limb tiredness) for this type of training (HIIT) led to an increase in SF-36 MCS scores with time [4,8,36,37]. QoL scores did not alter considerably, according to the findings of Candemir I et al. [32]. The number of physical function domains increased (SF-36, SGRQ activity domain, and PCS). When HRQoL scores in LTx were compared with normal, the greatest disadvantage was observed in the variation of physical health rather than any other mental health domain [29].

### 4.1. Telerehabilitation in Pulmonary Rehabilitation

Restrictions imposed due to the COVID-19 pandemic have had a negative impact on the healthcare system, including the rehabilitation of patients [38]. Telerehabilitation, as with any other technology, the primary problem is always accessibility, e.g., no stable internet connection, no smartphone app, no tablet/laptop, and no stable mobile internet access. Telerehabilitation is defined as the delivery of rehabilitation at a distance using a variety of information communication technologies, e.g., live video conferences, mobile apps, automated chatbots, etc. [38,39]. To date, little is known about the efficacy and effectiveness of telerehabilitation in lung transplant patients. However, there are some initial published results, e.g., an eight-week telerehabilitation programme was safe and effective, with increased patient exercise capacity, physical activity, lower limb strength, and improved balance [9,10,40]. Preoperative telerehabilitation is multimodal and has been shown to reduce postoperative complications and accelerate recovery, thus reducing morbidity and hospitalization [41]. Post-operative telerehabilitation in terms of physical activity can be varied, but in general, it always depends on the procedure and the region affected. Physiotherapy aims to fully restore range of motion and muscle imbalance, and an important adjunct is individualising nutritional therapy and psychotherapy, resulting in improved QoL for patients [9,41,42].

One of the undesirable effects of the COVID-19 pandemic is increased passive time at home, resulting in decreased cardiopulmonary capacity, muscle mass, and muscle strength, whereas regular exercise and physical activity significantly reduce fatigue and psychosocial dysfunction while increasing aerobic fitness, optimising body composition and QoL [43]. Resistance exercise increases contractile protein mass, reduces sarcopenia and body fat in rehabilitation participants, and increases muscle strength; sub-intensive aerobic exercise increases the mitochondrial number and oxidative enzyme activity in skeletal muscles and develops more efficient mitochondrial regulation in the body [44]. A number of studies suggest [10,13,41,45] that telerehabilitation is cost-effective and associated with patient satisfaction and adherence, but there are still underdeveloped components of telerehabilitation that need to be improved. Nevertheless, telerehabilitation is a very promising option for patient care, not only during the pandemic but also as a maintenance part of normal rehabilitation based on face-to-face encounters. 

### 4.2. Limitations of the Study

Our review was a review and not a systematic review of the literature, so studies may have been omitted. We focused primarily on RCT studies in English, so publications in other languages may have been omitted from the review. Due to manuscript length limitations, conference abstracts and meta-analyses were not included in this mini-review. Focusing on the exception of pre-transplant 6MWT, there is a lot of variation in the studies’ descriptions of the rehabilitation programmes, as well as what is measured afterward and when. There is probably a selection bias (because of the different physical conditions of the patients or who are being entered into the outpatient versus inpatient programmes).

## 5. Conclusions

In conclusion, rehabilitation programmes seem to be advantageous to patients both before and after lung transplantation. To achieve gains in exercise capacity, dyspnoea, and HRQoL, appropriate exercise parameters in terms of intensity, frequency, and duration are unavoidable. It is important to conduct more studies on the immediate and long-term impact of physical activity and/or training on survival and mortality, transplant rejection, risk, hypertension, infection rates, the onset of diabetes and/or obesity, and other effects on QOL.

## Figures and Tables

**Figure 1 life-13-00506-f001:**
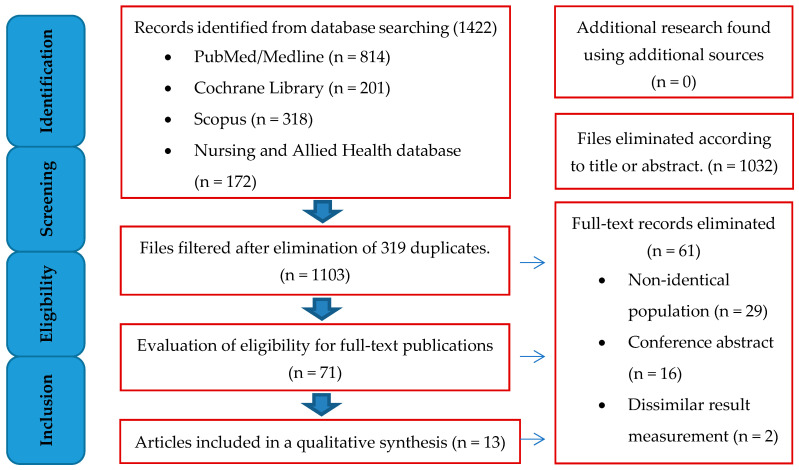
PRISMA flow diagram for articles involved in selection process.

**Table 1 life-13-00506-t001:** Pretransplant rehabilitation studies.

Study and Author	Patient Characteristics	Rehabilitation Programme	Measurements
The effectiveness of pulmonary rehabilitation in connection with lung transplantation in Hungary. Kerti M et al. [2]	▪ 63 participated in the pre-transplant rehabilitation group ▪ Age 58 ± 6.6 years ▪ Male/female: 34/29	▪ Breathing techniques ▪ Chest wall stretching ▪ Chest mobilization ▪ Muscle force ▪ One session/day, length: 30 min ▪ Three times/day: bicycle or treadmill; 15–20 min continuous or interval form/1 month.	▪ Test of pulmonary function ▪ A six-minute walk ▪ Test for chest wall enlargement ▪ Test of maximum inspiratory pressure ▪ Hand grip strength ▪ Breath-holding time test ▪ mMRC; CAT ▪ The BODE index is used to measure COPD severity.
Results	▪ Chest flexibility ▪ Training tolerance and QoL. ▪ Considerable modifications in CWE, CAT, BHT, FEV1%pred, and FVC% pred (*p* < 0.05)		
Study type	RCT		
Population	▪ 63 candidates for Ltx (40 COPD, 18 IPF, 4 bronchiectasis, 1 alveolitis fibrotisans); pre-transplant rehab programme and 14 post-transplant rehab programmes (9 COPD, 4 IPF)		
Effect of an 8-week Pulmonary Rehabilitation Program on Dyspnoea and Functional Capacity of Patients on Waiting List for Lung Transplantation. Kılıç L et al. [3]	▪ 23 patients ▪ Male/female: 13/10 ▪ Age: 35 ± 10 years	▪ Breathing exercises (local expansion exercises, diaphragmatic movements, and pursed lip breathing). ▪ Free exercise strolling ▪ Power workouts for the upper and lower extremities with TheraBands. ▪ Two 1 h sessions per week for 8 weeks (16 sessions). ▪ Additionally, a 3-day per week at-home training programme was proposed for all candidates.	▪ 6-min walk distance (6MWD). ▪ The Borg scale and the Medical Research Council (MRC) dyspnoea scale were used to measure the rate of dyspnoea.
Results	▪ A notable increase in 6MWD (median: 60 m; 360 [70–254] vs. 300 [139–489] m; *p* = 0.018). ▪ Clinical improvement in MRC and Borg scores (*p* = 0.008 and 0.0001, respectively).		
Study type	RCT		
Population	▪ 23 candidates were found to have LTx bronchiectasis (n = 10.4%), silicosis (n = 7.3%), sarcoidosis (n = 2.9%), idiopathic pulmonary fibrosis (n = 1.4%), chronic obstructive pulmonary disease (n = 1.4%), other conditions (n = 2.9%).		
Interval versus continuous training in lung transplant candidates: a randomized trial. Gloeckl R et al. [4]	▪ 60 patients ▪ 53 ± 6 years ▪ Male/female: 28/32	▪ Education and therapeutic breathing. ▪ CT or IT 5 days each week. ▪ CT (n = 30, cycling at 60% of peak work rate) or interval training (n = 30, cycling at 100% peak work rate for 30 s at a time, followed by 30 s of rest) during a 3-week inpatient rehabilitation programme.)	▪ Modification of a 6-min walk distance
Results	The 6-min walking distance achieves similar improvements of 35, 29 m for IT and 36 43 m for CT in both groups. In pre-lung transplant COPD patients, interval training is associated with a lower burden of dyspnoea and with fewer unintentional breaks during exercise but it also results in comparable improvements in physical capacity.		
Study type	RCT		
Population	▪ 60 candidates for LTx on waiting list with COPD		
Pulmonary rehabilitation improves survival in patients with idiopathic pulmonary fibrosis undergoing lung transplantation. Florian J et al. [5]	▪ N = 89 ▪ Male/female: 57/32 ▪ Age: 55.93 ± 10.93 years	▪ Breathing exercises (respiratory cycle) ▪ Strengthening the arms and legs with an initial charge of 30% of the maximum number of repetitions tested, followed by 1 set of 10 repetitions for each exercise. ▪ Treadmill aerobic exercises: starting at 70% of 6MWD test speed, adding 6 min at a time until 30 min were attempted. Every seven sessions, the speed was likewise increased by 0.3 km/h.	▪ Survival following LTx ▪ Days spent in the IMV, LOS in the ICU, and LOS in the hospital after LTx. ▪ Pulmonary function tests ▪ 6MWD test
Results	The Kaplan–Meier curve showed a significant rise in 6MWD (43 86 m, *p* = 0.005, and *p* = 0.002) Less time spent in the ICU (5 days vs. 7 days, *p* = 0.004) and less time spent in hospital (20 days vs. 25 days, *p* = 0.046) for the PRP group compared to the control group. Lower ICU death rate (*p* = 0.006) and a higher 5-year survival rate (89.9% vs. 60.9%, *p* < 0.001) for the PRP group.		
Study type	Quasi-experimental study		
Population	▪ 89 IPF patients were selected for unilateral LTx out of 464 contiguous candidates for LTx (278 underwent surgery).		
Nordic Walking in Pulmonary Rehabilitation of Patients Referred for Lung Transplantation. Ochman M et al. [6]	▪ Intervention group (n = 22) ▪ Age: 50.4 ± 7.84 years ▪ Sex: male/female: 22/0 ▪ Control group (n = 18) ▪ Age: 53.6 ± 8.79 years ▪ Sex: male/female: 16/2	▪ 12-week lung rehabilitation protocol of NW. ▪ Two cycles, 2 weeks in hospital, and 4 weeks at home.	▪ Quality of life ▪ Spirometry ▪ 6MWD ▪ Severity of Dyspnoea using Baseline Dyspnoea Index (BDI) questionnaire.
Results	At the 6-week mark, 11 patients with NW experienced a substantial improvement in their average 6MWD (from 310 to 364 m) *p* = 0.022 NW 12-week follow-up: 14 patients improved their 6MWT average distance to 374 m Control group follow-up: after 12 weeks, there was a statistically significant drop in the average 6MWT distance of the control group (from 326 to 268 m; *p* = 0.005). Forced vital capacity (FVC; from 47.66 to 52.78% expected, *p* = 0.009) and forced expiratory volume in 1 s (FEV1) improved, resulting in a change in the mean dyspnoea score on the MRC questionnaire, during or prior to rehabilitation. FEV1 and FVC levels decreased, and the FEV1/FVC ratio was higher in the control group (*p* = 0.042).		
Study type	Quasi-experimental		
Population	▪ LTx waiting list for 40 (Experimental group: 22, Control:18) (7 COPD cases and 15 cases with an ILD diagnosis (7 [IPF], 2 cases of allergic alveolitis, 1 case of sarcoidosis, 1 case of histiocytosis, 1 case of silicosis, 2 cases of bronchiectasis, and 1 case of nonspecific interstitial pneumonia).		
Pulmonary Rehabilitation in Patients with Advanced Idiopathic Pulmonary Fibrosis Referred for Lung Transplantation. Fontoura FF et al. [7]	▪ N = 48 ▪ Male/female: 31/17 ▪ Age: 57.1 ± 9.7 years	▪ 1 h per session of strength and aerobic training on a treadmill. ▪ 20 to 30 min of aerobic exercise while jogging on a treadmill with a subjective load of 3/10 on the modified category Borg scale. ▪ After 2 weeks, the intensity was raised when performing >12 repetitions/set comfortably.	▪ Spirometry ▪ 6-min walk test ▪ HQoL ▪ Dyspnoea calculated by the mMRC scale every day
Results	▪ 6MWD, dyspnoea, and impression of lower limb strength improve considerably. ▪ Four major domains of HRQoL considerably improve (vitality, emotional role, mental health, and physical functioning.		
Study type	▪ Cohort study		
Population	▪ IPF patients referred to LTx (n = 48)		
Telerehabilitation for Lung Transplant Candidates and Recipients During the COVID-19 Pandemic: Program EvaluationLisa Wickerson et al. [9]	▪ 78 LTx candidates and 33 recipients Candidates ▪ Age 59 ± 12 years ▪ Male/female: 37/41 Recipients ▪ Age 58 ± 12 years ▪ Male/female: 20/13	▪ Aerobic and resistance training programme tailored to the individual At least three times a week ▪ 26 LTx candidates used a treadmill, with sessions increasing in mean duration (from 16 to 22 min, *p* = 0.002) ▪ A total of nine LTx recipients used a treadmill that increased in speed (from 1.9 to 2.7 mph; *p* = 0.003)	▪ Physical activity ▪ Self-efficacy For Exercise (SEE) scale ▪ Aerobic and resistance exercise volumes ▪ 6-min walk test ▪ Short Physical Performance Battery (SPPB)
Results	▪ 35/42 (83%) candidates agreed that the app helped prepare them for surgery ▪ 18/21 (85%) recipients found the app helpful in their self-recovery ▪ On the Rapid Assessment of Physical Activity questionnaire (RAPA), 57% of LTx candidates scored as active, which improved to 87% (*p* = 0.02; n = 0 23). Quadriceps weight increased.		
Study type	▪ 4-week telerehabilitation programme		
Population	▪ 78 LTx candidates and 33 recipients		
Telerehabilitation Using Fitness Application in Patients with Severe Cystic Fibrosis Awaiting Lung Transplant: A Pilot Study Aimee M Layton et al. [10]	▪ Home PR (n = 11) ▪ Male/female: 6/5 ▪ age 30 ± 10 years ▪ Outpatient hospital PR (n = 8) ▪ Male/female: 0/8 ▪ age 29 ± 7 years	▪ Participants were provided with a personalised exercise programme and equipment, including a fitness application that provided exercise videos, recorded exercise time, and corresponding heart rate.	▪ Physical activity ▪ 6-min walk test ▪ Basic spirometry ▪ Cardiopulmonary Exercise Test (CPET) ▪ Borg scale
Results	▪ The main outcome was adherence. Secondary outcomes were adverse events. ▪ Completers of the home-based programme demonstrated a clinically meaningful lower decline in six MWD than noncompleters (six MWD −7 ± 15 vs. −86 ± 108 m).		
Study type	▪ Home-based pulmonary rehabilitation (PR) programme		
Population	▪ 19 CF patients		

BHT = Breath Holding Time; CF = Cystic Fibrosis; COPD = Chronic Obstructive Pulmonary Disease; CWE = Chest wall expansion test; FEV1 = Forced Expiratory Volume in one Second; FVC = Forced Vital Capacity; CPET = Cardiopulmonary Exercise Test; RAPA = Rapid Assessment of Physical Activity questionnaire; HRQoL = Health-related Quality of Life; LTx = Lung Transplantation; PR = Pulmonary rehabilitation; Qol = Quality of Life; RCT = Randomized Controlled Trial; 6MWD = Six-minute Walk Distance; 6MWT = Six-minute Walk Test; CAT = COPD Assessment Test); mMRC = Modified Medical Research Council; BDI = Baseline Dyspnoea Index; SEE = Self-efficacy For Exercise; SPPB = Short Physical Performance Battery.

**Table 2 life-13-00506-t002:** Post-transplant rehabilitation studies.

Study and Author	Patient Characteristics	Rehabilitation Programme	Measurements
The effectiveness of pulmonary rehabilitation in connection with lung transplantation in Hungary. Kerti M et al. [2]	▪ N = 14 ▪ Age 52 ± 8.8 ▪ Male/female: 11/3	▪ Breathing techniques ▪ Chest wall stretching ▪ Chest mobility ▪ Muscle force ▪ One session/day, length: 30 min ▪ 3 times/day: bicycle or treadmill: 15–20 min continuous or interval form/1 month	▪ Pulmonary function test: forced expiratory volume in one second (FEV1); Forced vital capacity (FVC) ▪ Chest wall expansion test (CWE); 6-min walk distance (6MWD) ▪ Test of maximum inspiratory pressure (MIP) ▪ Breath-holding test (BHT) ▪ Hand grip power (HGS) ▪ CAT measures QoL (COPD Assessment Test) ▪ COPD severity determined by the BODE-index and the mMRC (Modified Medical Research Council)
Results	Chest flexibility Training tolerance and quality of life (QoL). In the pre-transplant group, the severity of COPD decreased in terms of FEV1%pred, FVC%pred, mMRC, BHT, and HGS, but not in a significant way. Considerable modifications in CWE, CAT, BHT, FEV1%pred, and FVC%pred (*p*<0.05) The 6MWD, mMRC, and HGS did not significantly change.		
Study type	Randomized control trial (RCT)		
Population	14 LTx recipients		
Effect of high-intensity training on peak oxygen uptake and muscular strength after lung transplantation: A randomized controlled trial. Ulvestad M et al. [8]	▪ N = 25 ▪ Age: 52.3 ± 11.9 years ▪ Male/female: 11/14 ▪ CG (n = 29) ▪ Age: 51.1 ± 13.5 years ▪ Male/female: 16/13	▪ Mixed-mode High-intensity interval training (HIIT), strength conditioning, and endurance sessions 3 times/week. 20 weeks	▪ Cardiopulmonary examination to evaluate changes in VO_2_peak. ▪ The 1 maximum repetition (1RM) for the leg and arm presses differs. ▪ 36-Item Short-Form Health Survey ▪ Lung health (forced expiratory volume in 1 s, diffusing capacity of carbon monoxide). ▪ Physical agility (1RM in handgrip, 15-s stair running, and 30 s chair standing).
Results	▪ The between-group differences for 1RM arm press and leg press and mental aspect of SF-36 were 4.9 kg (95% CI = −0.1, 9.9) (*p* = 0.05), 11.6 kg (95% CI = 0.1, 23.0) (*p* < 0.05), and 5.7 kg (95% CI = 0.9, 10.4) (*p* = 0.02), respectively. ▪ When excluding participants with an attendance of <70% (n = 16), the between-group difference for VO2peak was 1.2 mL/(kg.min) (95% CI = 0.1, 2.4) (*p* = 0.032).		
Study type	▪ RCT		
Population	▪ 54 LTx recipients accordingly to 60 months following lung transplantation		
The Efficacy of Outpatient Pulmonary Rehabilitation after Bilateral Lung Transplantation. Candemir I et al. [11]	▪ N = 23 ▪ Completed PR (n = 17) ▪ Male/female 15/2 ▪ Age 47 ± 10.51 years ▪ Did Not Complete PR (n = 6) ▪ Male/female 5/1 ▪ Age 42 ± 9. 42 years	▪ 8 weeks in-hospital multidisciplinary outpatient rehabilitation programme. 2 months after LTx operation	▪ Battery testing for the incremental and endurance shuttling walks ▪ Power in the hands and quadriceps ▪ The maximum inspiratory and expiratory pressures of the respiratory muscles ▪ Mental health state and QoL ▪ The distance obtained using the ISWT, and ESWT is used to evaluate exercise function. ▪ For the ESWT, the incremental shuttle test was used to measure VO2peak, and the speed frequency of walking was then calculated using 85% of VO2peak.
Results	▪ Notable variations in MEP measurement results. ▪ Anxiety scores are significantly different (*p* < 0.05). ▪ The ISWT distance before PR (23%) and anticipated to (36%) after PR differed (*p* < 0.001). ▪ After PR, ISWT and ESWT levels increased (*p* < 0.05). ▪ After PR, the average quadriceps score increased with a statistically significant difference (*p* < 0.001). ▪ A statistically significant difference was also observed in the early post-operative period following LTx in training capacity, skeletal muscle, diaphragm muscle strength, body composition, QoL, and psychological health status.		
Study type	▪ Quasi-experimental study		
Population	▪ 23 LTx recipients: COPD (30%), IPF (30%), histiocytosis-X (13%), silicosis (9%), bronchiectasis (9%), α-1 antitrypsin deficiency (4%), and pulmonary alveolar proteinosis (4%).		
Improvements in functional and cognitive status following short-term pulmonary rehabilitation in COPD lung transplant recipients: a pilot study. Andrianopoulos V et al. [12]	▪ N = 24 ▪ Male/female: 14/10 ▪ Age 58.2 ± 6.3 years	▪ PR protocol: in-patient exercise on treadmills, stationary bikes, and strength conditioning equipment ▪ Lifting weights while performing workouts for the upper and lower limbs training is performed gradually until dyspnoea and leg fatigue scores on the Borg scale reach >6/10.	▪ Anthropometric measurements, pulmonary function exploration, and blood gas exchange analysis. ▪ Mental health questionnaires (Hospital Anxiety and Depression) Scale (HADS). ▪ Chronic Respiratory sickness Questionnaire (CRQ). ▪ 6-min walk test (6MWT) ▪ Cognitive assessment using a battery of neuropsychological tests based on the Intelligence Quotient test (IQ) and Stroop colour-word test
Results	▪ A statistically significant difference in respiratory function based on diffusion capacity for carbon monoxide (+4.3%; *p* = 0.012) and static hyperinflation (residual volume/total lung capacity −2.3%; *p* = 0.017). ▪ Statistically significant difference (6-min walk test +86 m; *p* < 0.001). ▪ A statistically significant difference (ES latitude 0.23–1.00; all *p* ≤ 0.34) in 50% of the mental examinations. ▪ Memory capacity and learning capabilities (ES scores 0.62 and 0.31, respectively).		
Study type	▪ Pilot study		
Population	▪ 25 LTx recipients (1 month after operation)		
Delivering an in-Home Exercise Program via Telerehabilitation: A Pilot Study Jiyeon Choi [13]	▪ N = 4 ▪ Patient 1, Male 66 years ▪ Patient 2, Male 62 years ▪ Patient 3, Male 62 years ▪ Patient 4, Female 30 years	▪ 8-week in-home exercise intervention for lung transplant recipients using a telerehabilitation platform ▪ aerobic and strengthening exercises	▪ SenseWear Armband^®^ was used to measure 7-day physical activity ▪ 6-min walk distance ▪ Berg balance scale ▪ 30 s chair stand test ▪ SpO_2_
Results	▪ Participants improved walking distance (6-min walk distance), balance (Berg Balance Scale), lower body strength (30 s chair stand test), and steps walked (SenseWear Armband^®^) ▪ Of importance, improvement in 6MWD for three of four participants exceeded the minimal clinically important difference (54 m).		
Study type	▪ Pilot Study of Lung Transplant Go (LTGO)		
Population	▪ Four patients (three IPF + one CF)		

BHT = Breath Holding Time; CF = Cystic Fibrosis; CG = Control Group; COPD = Chronic Obstructive Pulmonary Disease; CI confidence interval; CWE = Chest wall expansion test; FEV1 = Forced Expiratory Volume in one Second; FVC = Forced Vital Capacity; HGS = Hand Grip Strength; HIIT = High-Intensity Interval Training; HRQoL = Health-related Quality of Life; IIP = Idiopathic Interstitial Pneumonia; IQ = Intelligence Quotient; LTx = Lung Transplantation; LTGO = Lung Transplant Go; PR = Pulmonary rehabilitation; Qol = Quality of Life; RCT = Randomized Controlled Trial; 6MWD = Six-minute Walk Distance; 6MWT = Six-minute Walk Test; SpO2 = Oxygen saturation; HADS = Hospital Anxiety and Depression Scale; CRQ = Chronic Respiratory sickness Questionnaire; MIP = maximum inspiratory pressure; CAT = COPD Assessment Test); mMRC = Modified Medical Research Council; 1 RM = 1 maximum repetition; SF-36 = 36-Item Short-Form Health Survey; ESWT = endurance shuttle walk test; ISWT = incremental shuttle walk test; VO2 = oxygen consumption.

## Data Availability

Not applicable.

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
