# Peer review of "Lung Transplant Rehabilitation—A Review"

_life, 2023, doi:10.3390/life13020506_

Round 1

Reviewer 1 Report (Previous Reviewer 1)

The manuscript reads a lot better in light of the revisions which have been undertaken. However, it does require a bit more work.

1) The results section in the abstract does not really encapsulate the findings from this review. You need to populate this section with data indicating how many of the manuscripts of interest contained data where there were statistically significant findings obtained for the various outcomes of interest. Otherwise, all you have here is a results section made up of sweeping statements-which does not reflect the findings of this particular review

2) The results section requires a bit more work. If you are not going to include data on the number of subjects in each study, along with their age and gender in the Tables, then the first paragraph of the results text for each of the subsections- Pretransplant rehabilitation study/Post transplant rehabilitation study-does. Currently what is presented is inadequate in the first paragraph of each of these two subsections. You need to insert summary data elements beside each of the percentages at a minimum

3) Plus, it is essential not to replicate what is contained in the Tables also in the text of the results section. If anything, the text in the Tables particularly for the outcomes/results, can be condensed down further (because it is fairly wordy). That is if there is no significant difference in outcomes for a particular result then this can be stated outright.

4) The discussion section needs more work particularly the paragraph on limitations of this study. It does appear that there is a reasonable amount of heterogeneity between the studies of in how the rehabilitation programmes are both structured and managed, along with what is then measured (except for the 6MWDT pretransplant), and at what time points. Plus, there is more likely than not a selection bias with respect to which patients enter these research programmes (that is that it is possible that it is the patients who are more physically robust who are being entered into the outpatient rehab programmes compared to the patients who are being enrolled into the inpatient programmes). This is a missing piece of the puzzle so to speak.

5) What is meant by the term quasi experimental study? This term appears both in the Table as well as elsewhere in the manuscript

6) In the conclusions section at the end of the discussion section mention is made that it is important to "follow the main guidelines for training interventions in pulmonary rehabilitation programmes" Whose guidelines are you referring to (the only mention made of guidelines in the discussion section is those from British Thoracic Society-and they seem problematic)? Do any of the international Transplant organizations have guidelines for this?

Author Response

1) The results section in the abstract does not really encapsulate the findings from this review. You need to populate this section with data indicating how many of the manuscripts of interest contained data where there were statistically significant findings obtained for the various outcomes of interest. Otherwise, all you have here is a results section made up of sweeping statements-which does not reflect the findings of this particular review

Answer: Thank You very much for your comment. The significant results of the 13 selected studies are described in detail in the tables and results. Unfortunately, we cannot list as much data in the abstract.

2) The results section requires a bit more work. If you are not going to include data on the number of subjects in each study, along with their age and gender in the Tables, then the first paragraph of the results text for each of the subsections- Pretransplant rehabilitation study/Post transplant rehabilitation study-does. Currently what is presented is inadequate in the first paragraph of each of these two subsections. You need to insert summary data elements beside each of the percentages at a minimum.

Answer: Thank You very much for your valuable comment, the tables have been checked and all studies now include the number, age and sex of participants, so some of the results were not repeated again.

3) Plus, it is essential not to replicate what is contained in the Tables also in the text of the results section. If anything, the text in the Tables particularly for the outcomes/results, can be condensed down further (because it is fairly wordy). That is if there is no significant difference in outcomes for a particular result then this can be stated outright.

Answer: Thank you very much for your valuable comment, the tables have been condensed, the dead spaces have been minimized and removed what is not significant.

4) The discussion section needs more work, particularly the paragraph on limitations of this study. It does appear that there is a reasonable amount of heterogeneity between the studies of how the rehabilitation programmes are both structured and managed, along with what is then measured (except for the 6MWT pretransplant), and at what time points. Plus, there is more likely than not a selection bias with respect to which patients enter these research programmes (that is that it is possible that it is the patients who are more physically robust who are being entered into the outpatient rehab programmes compared to the patients who are being enrolled into the inpatient programmes). This is a missing piece of the puzzle so to speak.

Answer: Thank You, we have corrected the limitations of the study.

5) What is meant by the term quasi experimental study? This term appears both in the Table as well as elsewhere in the manuscript.

Answer: Thank You, the quasi-experiments are studies that aim to evaluate interventions but that do not use randomization. Similar to randomized trials, quasi-experiments aim to demonstrate causality between an intervention and an outcome.

6) In the conclusions section at the end of the discussion section mention is made that it is important to "follow the main guidelines for training interventions in pulmonary rehabilitation programmes" Whose guidelines are you referring to (the only mention made of guidelines in the discussion section is those from British Thoracic Society-and they seem problematic)? Do any of the international Transplant organizations have guidelines for this?

 Answer: Thank You for your valuable comment, this part has been deleted from the manuscript.

Reviewer 2 Report (Previous Reviewer 3)

Dear authors,

Congratulations on an exciting review of a crucial topic, especially in times of pandemic and teleconsultation.

I have no particular criticism to address of your work, and you already stated the limitations of this mini-review satisfactorily. As you have discussed, a limited number of small single-center studies indicate that outpatient rehabilitation involving supervised exercise training could be helpful for patients to improve clinically relevant outcomes in both the pre-transplant and post-transplant phases. However, this is only supported by low-quality evidence, and none of the existing RCTs measured the effects of exercise training on crucial long-term outcomes, such as sustained improvement in QOL and participation in daily activities, survival, the incidence of metabolic and cardiovascular morbidity, and cost-effectiveness. Cohort studies in this patient group offer limited information due to the considerable spontaneous improvements in outcomes generally observed in the immediate post-transplant phase.

I would ask you to make the text more fluent by eliminating some redundant statements.

Good job.

Author Response

  1. Congratulations on an exciting review of a crucial topic, especially in times of pandemic and teleconsultation. I have no particular criticism to address of your work, and you already stated the limitations of this mini-review satisfactorily. As you have discussed, a limited number of small single-center studies indicate that outpatient rehabilitation involving supervised exercise training could be helpful for patients to improve clinically relevant outcomes in both the pre-transplant and post-transplant phases. However, this is only supported by low-quality evidence, and none of the existing RCTs measured the effects of exercise training on crucial long-term outcomes, such as sustained improvement in QOL and participation in daily activities, survival, the incidence of metabolic and cardiovascular morbidity, and cost-effectiveness. Cohort studies in this patient group offer limited information due to the considerable spontaneous improvements in outcomes generally observed in the immediate post-transplant phase.

I would ask you to make the text more fluent by eliminating some redundant statements.

Good job.

Answer: Thank You very much, we have updated the manuscript.

Reviewer 3 Report (Previous Reviewer 4)

Please find attached comments for your consideration 

Author Response

I thank the journal for sending me the paper to present a reevaluation of a previous submission now presented as a systematic mini-review titled “Lung transplant rehabilitation: mini-review” by Abidi et al. Based on published literature survey, the authors have consolidated information on the effectiveness of training on exercise capacity and QoL in pre- as well as post-operative lung transplant patients. This mini review identify that rehabilitation programs appear to be beneficial for patients prior to and following lung transplantation. Some factors contributing to the improvement in exercise capacity, dyspnea and HRQoL were the time scale of training parameters, and frequency of work intensity. Overall, this article is informative and will add relevant information to the field. There are some suggestions that the authors are requested to consider/made amends before this paper can be accepted for publication –

  1. Line 85 – please be more specific – from the beginning = when?

Answer: Thank You for your valuable comments, we have corrected the manuscript.

  1. Lines 150-151 – is unclear, please revise. Pretransplant studies, the authors say 6 studies were included, but as per these lines 3 RCT + 2 quasi + 2 cohort studies + 1 pre-transplant study = 8 studies in all? Also, the last study – pre-transplant study (why is it a separate study in the group that is called “pre-transplant”? Table 1 then details 8 studies instead of 6.

Answer: Thank You for your valuable comment, we have corrected the text.

  1. Table 1 (8 studies here rather than 6 mentioned in text) presentation is difficult to read and has a lot of dead space within the table information, perhaps due to the pdf file formatting? Please ensure the information and table presentation is reformatted and made more compact. For example, please include the “study type” and “population” rows in the first column (“study and author”) itself to save space and minimize dead space. Table 2 – same criticism as above (in point 3). Please revise accordingly.

Answer: Thank You for your valuable comments, the tables have been formatted and dead spaces have been minimized.

  1. Discussion section – several places the grammar and sentences are poorly constructed– For example, Line 233 – “for patients with LTx” not LTx. Line 240 – enhancements in “life quality for patients with LTx” not LTx life quality. Line 246 – what do the authors mean by “his paper”? Line 261 – “Florian J are some of the few scientists” – language corrections mandated. Line 307/308 – what is “m”? Line 328/352 (and other places within paper) – use the “QoL”

  Answer: Thank You for your valuable comments, in discussion we have made corrections and inserted QoL in several places.

Round 2

Reviewer 1 Report (Previous Reviewer 1)

The manuscript reads better in light of the revisions which have been undertaken. One minor point. Can you please go through Table 1 and ensure that the patient demographics are all reported in the same manner for each of the studies-currently there is some variation in how the numbers and percentages are depicted. This needs to be addressed

Author Response

Thank You very much, I have corrected the demographic data everywhere. Yours sincerely, Janos Varga

This manuscript is a resubmission of an earlier submission. The following is a list of the peer review reports and author responses from that submission.

Round 1

Reviewer 1 Report

There is one significant problem with this manuscript and that is that it replicates the findings of a systematic review published in 2020 on this very same topic (which is not referenced)-

Exercise training for lung transplant candidates and recipients: a systematic review | European Respiratory Society (ersjournals.com)

Plus, the addition of your own units data towards the end of the results section is not consistent with the primary focus of undertaking a systematic review approach or reviewing the published literature

In some respects you are best to consider publishing your own units experience and then discussing the results based on the published literature-ie completely shifting your focus

Author Response

Thank You very much for your valuable comment, the text of the manuscript has been changed and the above mentioned article is cited for reference. Our own data has been permanently removed from the manuscript.

Reviewer 2 Report

Thanks for the opportunity to read this work. Although the idea and the planning are interesting, the paper is difficult to read, not very linear. Moreover, the Authors' experience is certainly interesting but unrelated to the work and should be insert in another way. It is not anticipated neither in the title nor in the Materials, and there is no resumption in the discussion. Alternatively, it is possible to modify the layout of the paper, such as the presentation of the results of the Authors' experience and the systematic review of the literature.

Author Response

Thank You for your valuable comment, we have changed the title and the text to make the manuscript easier to understand and read. The tables have also been thoroughly reworked.

Reviewer 3 Report

Dear authors, I read your study with interest. I always find the topic of rehabilitation in thoracic surgical patients, especially lung transplant patients, fascinating.

Unfortunately, I believe several problems must be solved to make your manuscript worthy of publication.

1) the title: systemic is not the best word you could have chosen. Neither is "systematic" unless you rewrite the study taking care of the guideline you cited (PRISMA); "a comprehensive review of the literature" sounds better, in my opinion.

2) in the introduction, I would reduce the part about reports on lung transplants per se, and go directly to the target: recovery, rehabilitation methods, and results.

3) in Table 1, I believe that reporting the title of each study may appear redundant. I suggest including only the lead author and leaving the reader the option to search for the study title among the references.

4) Paragraph "New data from our research group in lung transplant": your results are exciting. In the methods, you should reference your case history and the statistics used to obtain the significance. Reporting a mere report of your series makes the whole argument unscientific.

5) Reference list should be implemented with more recent papers

6) Comprehensive text revision for typos and extensive English language revision is suggested.

Author Response

Thank You very much for your valuable comments and your review.

1) Thank You, we have changed the title to Lung Transplant Rehabilitation - Mini-Review.

2) Thank You, the introduction has been rewritten and shortened.

3) Thank You, we felt it important to include the author and title in the tables, but of course if the Reviewer wishes, we will remove the titles from the tables.

4) Thank You, we have removed all new data from the manuscript.

5) Thank You, we have added new studies to the reference list.

6) Thank You, text has been rewritten and proofread by a foreign language teacher.

Reviewer 4 Report

Please find attached my comments. 

Author Response

  • Thank You for your valuable comment, one year as a limit is a classic in rehabilitation e.g. https://pubmed.ncbi.nlm.nih.gov/19481017/, https://pubmed.ncbi.nlm.nih.gov/33115788/ that's why we wrote 12 months.
  • Thank You, all clinical recommendations have been removed from the manuscript.
  • Thank You very much for your valuable comment, all figures and tables have been re-edited to make them easy to read and understand for everyone.
  • Thank You very much for your valuable comment, we have modified the methodology and added quality of life, exercise: "training exercise," "exercise training capacity," "quality of life"
  • Thank You very much for your valuable comments, all new data has been removed from the manuscript.
  • Thank You very much for your valuable comments, we have deleted lines 290-291.
  • Thank You very much, we have rewritten the discussio.
  • Thank You very much, we have deleted the clinical recommendations from the manuscript.
  • Thank You very much, we have corrected the grammatical errors in the manuscript.
